# Unhealthy Dietary Patterns and Their Associations with Sociodemographic Factors as Predictors among Underweight and Overweight Adolescents in Southern Thailand

**DOI:** 10.3390/ijerph20176703

**Published:** 2023-09-01

**Authors:** Pikuntip Kunset, Chuchard Punsawad, Rewwadee Petsirasan, Charuai Suwanbamrung, Shamarina Shohaimi, Udomsak Narkkul, Naiyana Noonil

**Affiliations:** 1Excellence Center of Community Health Promotion, School of Nursing, Walailak University, Nakhon Si Thammarat 80160, Thailand; pikuntip.kun@gmail.com; 2Department of Medical Science, School of Medicine, Walailak University, Nakhon Si Thammarat 80160, Thailand; chuchard.pu@wu.ac.th (C.P.); udomsak.na@wu.ac.th (U.N.); 3Faculty of Nursing, Prince of Songkla University, Songkhla 90110, Thailand; rewwadee.p@psu.ac.th; 4Excellent Center for Dengue and Community Public Health School of Public Health, Walailak University, Nakhon Si Thammarat 80160, Thailand; yincharuai@gmail.com; 5Department of Biology, Faculty of Science, University Putra Malaysia, Serdang 43400, Selangor, Malaysia; shamarina@upm.edu.my

**Keywords:** unhealthy dietary patterns, sociodemographic factors, adolescents

## Abstract

(1) Background: Adolescence is a critical developmental phase; dietary intake and nutritional status significantly impact health outcomes. (2) Objective: This cross-sectional study investigated dietary patterns (DPs) and the association between sociodemographic factors and unhealthy DPs among adolescents in Thailand. (3) Methods: A multi-stage sampling selected 1480 participants from three public high schools in Nakhon Si Thammarat province. A food frequency questionnaire assessed dietary habits, and principal component analysis was used to identify DPs. Multinomial logistic regression examined the association between sociodemographic factors and DPs. (4) Results: The findings show that 25.9% of adolescents were underweight, 14.7% were overweight, and 5.8% were obese. Three DPs were identified: a healthy ‘protein and vegetables’ pattern and two unhealthy patterns: ‘snacks’ and ‘processed foods’, which explained 12.49%, 10.37%, and 7.07% of the dietary variance, respectively. Among underweight adolescents, higher snack consumption was associated with being younger (odds ratio (OR) = 3.24) and receiving a higher daily allowance (OR = 3.43). Additionally, female adolescents who engaged in frequent exercise had a 2.15 times higher intake of processed foods. Among overweight adolescents, higher snack intake was linked to being younger (OR = 8.65) and having larger families (OR = 6.37). Moreover, an increased daily allowance was associated with higher consumption of processed foods (OR = 11.47). (5) Conclusion: This study underscores the socio-demographic influence on unhealthy DPs. Insights can guide targeted interventions to foster healthier dietary habits during adolescence.

## 1. Introduction

Adolescence is a critical phase characterized by rapid growth and development, during which individuals have increased nutritional requirements. This makes them more susceptible to malnutrition, seriously affecting their health and well-being [1]. Underweight adolescents may experience delayed maturation, reduced muscle strength, and limited physical work capacity. They may also have lower bone density, which can increase their risk of fractures later in life [2]. Obesity in adolescents is associated with a higher risk of early-onset chronic conditions, such as type 2 diabetes, hypertension, and cardiovascular disease [3]. Malnutrition poses a significant public health concern, particularly in lower- and middle-income countries [4]. A study of 57 low- and middle-income countries found that the Southeast Asia region had the highest prevalence of underweight (17.9%) and the lowest prevalence of overweight (11.1%) [4]. However, according to the 2019 Global School-based Student Health Survey (GSHS) conducted by the World Health Organization (WHO), the prevalence of those underweight among Thai adolescents aged 13–17 years was reported at 8.4%; the overall prevalence of overweight and obesity in the same age group was 18.9% and 6.6%, respectively [5]. The Thai government has implemented a policy addressing childhood obesity, with a target of limiting its prevalence to no more than 10% [6]. The WHO has also provided a priority goal to reduce the prevalence of obese adolescents through obesogenic environment management, including the promotion of healthy foods and physical activity, improving a healthy school environment and nutrition health literacy, and providing healthy habits in the family [7].

The WHO recommends focusing on dietary patterns as the preferred approach for nutritional epidemiology studies. This is because healthy development depends not solely on a single food or nutrient but on the combination of foods consumed [7]. Maintaining a healthy dietary intake from adolescence to adulthood is crucial for sustaining these positive behaviors [8]. Many researchers have examined dietary patterns (DPs) among adolescents in different countries, recognizing that food choices often involve the combination of various foods [9,10,11]. Adolescents are particularly susceptible to unhealthy eating behaviors, such as snacking, skipping meals, eating out, eating late, and consuming fast foods, and do so more frequently than younger children [12,13]. Additionally, sociodemographic factors, such as age, sex, family structure, and income, and health-related behaviors, such as smoking, drinking, and lack of physical activity, can contribute to an unhealthy diet, often starting during adolescence and influenced by nutritional factors [13,14]. Therefore, analyzing dietary patterns can provide a more comprehensive understanding of eating habits and shed light on the complexity of dietary intake [15].

Over the past two decades, Thai adolescents have shifted from traditional rice and vegetable-based diets to unhealthy eating patterns characterized by high fat, salt, and sugar but low fiber content [16]. This dietary transition has been primarily driven by the easy accessibility of street food, such as the snacks, sugary drinks, and highly energy-dense items sold at tuck shops and convenience stores near schools or along school routes [17]. Despite this concerning dietary trend, research on the dietary patterns of Thai adolescents remains limited. In this study, we hypothesize that sociodemographic factors are associated with an unhealthy diet based on nutritional status. Therefore, this study aims to assess Thai adolescents’ dietary patterns and investigate the sociodemographic factors associated with unhealthy dietary patterns among underweight or overweight adolescents.

## 2. Materials and Methods

### 2.1. Study Design and Setting

This analytic cross-sectional study was conducted among adolescents from January to April 2021, focusing on their sociodemographic factors, nutritional status, and food frequency. Using random sampling, the study was conducted in three public high schools in the Thasala, Maung, and Cha-uat districts of Nakhon Si Thammarat province in southern Thailand, divided based on the most significant number of students. Nakhon Si Thammarat province is approximately 780 km from Bangkok and encompasses 23 districts. This study randomly selected three districts representing the northern, central, and southern regions of Nakhon Si Thammarat province which cover all regions based on several geography of this province (Figure 1). This province is characterized by a vast land area and high population density, making it the largest part of southern Thailand. Moreover, its diverse topography contributes to the availability of a wide variety of local produce, including rice, fruits, vegetables, and seafood.

### 2.2. Participants

The study specifically included students in grades 10 to 12 enrolled in the first semester of the 2020 academic year at public high schools. Students who were ill or injured in a way that could affect their dietary habits or nutritional status, had food restrictions, or had incomplete food assessments were excluded from the study.

### 2.3. Sample Size and Sampling Technique

The population of this study comprised 52,254 students enrolled in public high schools located in Nakhon Si Thammarat province. The sample size was calculated using G*power, considering the z-test and logistic regression as statistical tests. An a priori power analysis was conducted, considering an odds ratio of 2.047 [18], a significance level of 0.05, and a power of 1 − β = 0.95. This resulted in a sample size of 1480 adolescents, with an additional expected dropout rate of 10%.

A multi-stage sampling technique was employed to obtain a representative sample for the study. First, three public high schools with the highest student enrollment were selected from the Nakhon Si Thammarat province’s Thasala, Maung, and Cha-uat districts. Second, students were chosen from each school based on their enrollment numbers. Finally, students were randomly selected on the survey day, and their sociodemographic data, anthropometric measurement, and food intake frequency were collected. Thus, the sample selection was based on proportional sampling and inclusion criteria.

### 2.4. Research Tools

#### 2.4.1. Sociodemographic Data

Sociodemographic information was collected through a self-administered questionnaire consisting of two parts. The first part gathered participants’ data, including sex, age, and a cumulative grade point average (GPA), categorized as ≤3.00 vs. >3.00, where the national average GPA is 3.00. The questionnaire also included questions about daily pocket money (≤100 THB vs. >100 THB per day, equivalent to around 3 USD at the time of the study when the exchange rate was approximately 35 THB per dollar). Health-related behaviors were assessed, such as smoking status (current smoking: yes vs. no), exercise frequency (at least 30 min per day for ≤3 days/week vs. >3 days/week), and breakfast intake frequency, which was categorized as consuming breakfast ≤ 3 days/week vs. >3 days/week. The second part of the questionnaire focused on family data, including the number of family members (small: ≤5 members vs. large: >5 members), the primary caregiver or guardian of the participant, the person responsible for cooking in the family, the education level of the guardian, and the average family income.

#### 2.4.2. Nutritional Status

Anthropometric measurements, including weight and height, were taken to assess the nutritional status of the participants. Weight was measured using a calibrated digital scale (Tanita) with an accuracy of 0.1 kg, and height was measured using a stadiometer to the nearest 0.1 cm. The research team received training from expert nutritionists in anthropometry to ensure measurement accuracy and reliability. The nutritional status of the participants was categorized using the body mass index (BMI-for-age), which takes into account sex, date of birth, weight, and height. Z-scores were analyzed using the WHO Anthro Plus software for children and adolescents aged 5–19. This software divides nutritional status into five categories, including severe thinness (SD more than −3 SD), thinness (between −2 to −3 SD), normal weight (between −2 to +1 SD), overweight (between +1 to +2 SD), obesity (between +2 to +3 SD), and severe obesity (more than 3 SD) [19]. The nutritional categories in this study are underweight (thin: −2 SD and severe thin: −3 SD), normal weight (normal: −1 to +1 SD), and overweight (overweight: more than +1 SD, obesity: +2 SD, and severe obesity: +3 SD).

#### 2.4.3. Dietary Assessment and Food Groupings

The participants in this study completed a validated food frequency questionnaire (FFQ) to evaluate their dietary intake over the previous month. The FFQ was modified from the Thai food consumption survey [17], which was adapted by selecting and combining food items into food groups. The FFQ consisted of 49 food items categorized into 21 food groups, including rice, instant noodles, noodles, snacks, bakeries, sweetened drinks, soda drinks, Thai desserts, fish, chicken, pork, eggs, sweetened milk, milk products, soy milk, beans, processed meat, stir-fried meat with vegetables, Thai dishes, fresh vegetables, and fruits (Table 1).

The questionnaire was administered by the primary investigator, who received training from expert nutritionists. For each food item, participants were asked to indicate the frequency of their consumption over the previous month, with response options ranging from never eating (scored as 0), consuming less than 1 time per month (scored as 1), eating 2–3 times per month (score as 2), eating 1–3 times per week (score as 3), eating 4–6 times per week (score as 4) to eating every day (scored as 5). The FFQ demonstrated good content validity with a content validity index (CVI) of 0.89. Additionally, the FFQ was provided with 30 similar participants for the pilot study, and its reliability was assessed using Cronbach’s alpha coefficient, which yielded a value of 0.75.

### 2.5. Statistical Analysis

Descriptive statistics were used to report means and standard deviations (SD) for continuous variables, while frequencies (n) and percentages (%) were used for categorical variables. The chi-square test assessed differences in sociodemographic variables across the three nutritional status groups (underweight, normal weight, and overweight with obesity).

Principal component analysis (PCA) is a statistical analysis technique that can be used to reduce a large number of variables to a smaller number of components that account for most of the variance in the original variables set. The first component combines the maximum amount of the total variance, and the second component extracts the most significant variance not accounted for by the first component. This is correlated with [20].

In this study, PCA was used to identify dietary patterns. The Varimax orthogonal rotation method was used to analyze factor loading and establish pattern correlations. The adequacy of the sample was evaluated using the Kaiser–Meyer–Olkin test (KMO > 0.5), and the suitability of the data for factor analysis was assessed using the Bartlett test of sphericity (BTS) (*p* < 0.05). The scree plot was examined to determine the number of components with an eigenvalue > 1.0. Communalities were assessed, with a minimum value of 0.10 considered sufficient to explain the factor. Food groups with a factor loading > 0.25 [11,21] were identified as significant associations with the factor. The dietary patterns (DPs) were named based on the food items with the highest factor loading. Factor scores were calculated for each participant using the regression scoring method. Quartile cut points (Q1–Q4) were determined for each DP based on the participants’ consumption, where Q1 represented the lowest consumption, and Q4 represented the highest consumption of the respective DP.

Multinomial logistic regression analysis was used to examine the relationship between potential sociodemographic variables and Q2–Q3 of the DPs, with Q1 as the reference category compared with Q2–Q4. Adolescents who were normal weight (*n* = 761) were excluded from the regression model. Assumptions were tested, including that the dependent variable was of a nominal level, that the independent variable was continuous, and that there was no multicollinearity. Adjusted odds ratios (AOR) and 95% confidence interval (CI) were reported. All statistical tests were two-tailed, with a significance level of *p* < 0.05.

### 2.6. Ethical Consideration

The study protocol was reviewed and approved by the Human Research Ethics Committee of Walailak University (no. WUEC-20-352-01). As most participants were minors, the research team obtained permission from school principals and teachers to provide written informed consent forms to the participant’s parents or guardians. The participants were told their participation was voluntary and that they could withdraw from the study anytime. Verbal consent was obtained from the participants, and their parents or guardians signed the informed consent forms before data collection. This was done to protect the rights and welfare of the participants during the study.

## 3. Results

### 3.1. Characteristics of Participants

One thousand four hundred fifteen adolescents aged 13–18 participated in this study, averaging 15 ± 1.6 years. The participants were in grades 7 to 12. Most participants (53.8%) were of normal weight, while 25.7% were underweight, 14.7% were overweight, and 5.8% were obese. Females had slightly higher percentages in the normal weight category. Regarding daily pocket money, most participants (82.8%) received 100 THB/day or less. Furthermore, 12.9% reported being smokers. Interestingly, among the participants who were current smokers, 34.3% were underweight, and 19.7% were overweight. Most participants lived with small families (80.9%), and their mothers were responsible for preparing their meals (92.7%). Significant differences were observed in the participants’ sociodemographic characteristics across the three nutritional status groups, with sex and current smoking status being the most prominent factors (Table 2).

### 3.2. Dietary Patterns

The data obtained in this study met the necessary prerequisites for principal component analysis (PCA), as indicated by a Kaiser–Meyer–Olkin (KMO) measure of sampling adequacy of 0.827 and a significant *p*-value of <0.001 in Bartlett’s test of sphericity (BTS). The Chi-square value for the test was 4150.167. The factor loading of the DPs derived from the PCA of the self-reported FFQ completed by Thai adolescents is presented in Figure 2. The analysis identified three distinct DPs. The first DP, labelled ‘protein and vegetable’, showed a positive association with the consumption of milk products, soy milk, fresh vegetables, stir-fried meat with vegetables, eggs, beans, pork, Thai dishes, and sweetened milk in descending order of PCA loading. The remaining two patterns were categorized as unhealthy. The second DP, named ‘snacks’, is positively associated with Thai desserts, salty snacks, bakeries, and rice. The third DP, ‘processed foods’, was characterized by a positive loading of instant noodles, processed meat, soda drinks, noodles, and chicken. These results indicate that Thai adolescents’ DPs comprise a combination of healthy and unhealthy food choices, with some relying on fast food and snacks.

### 3.3. Dietary Patterns and Associated Factors

This study examined the adjusted odds ratios of sociodemographic factors related to dietary patterns among underweight and overweight adolescents. A comparison was made between the lowest quartile (Q1) and the higher quartiles (Q2–Q4). The analysis revealed that no significant factors were associated with the consumption of healthy ‘protein and vegetables’ patterns in either the underweight or overweight group. However, specific sociodemographic factors were linked to the consumption of unhealthy DPs, namely the ‘snack pattern’ and ‘processed foods’.

Among underweight adolescents, it was found that higher consumption of the snack pattern was associated with younger age (AOR: 3.24, 95%CI: 1.55–6.76). Additionally, receiving higher daily pocket money was associated with an increased intake of these foods (AOR: 3.43, 95%CI: 1.25–9.39). The study also revealed that frequent exercise significantly influenced a higher intake of processed foods (AOR: 2.15, 95%CI: 1.04–4.46). Furthermore, only 22% of males had the highest consumption of processed foods, indicating a lower likelihood than females (AOR 0.22, 95%CI: 0.10–0.47). Moreover, a higher frequency of breakfast consumption was associated with lower consumption of processed foods, with 37% lower odds of consumption (AOR: 0.37, 95%CI: 0.17–0.81) (Table 3).

The study also found that among overweight and obese adolescents, the highest consumption of the snack pattern was associated with being younger (AOR: 8.65, 95%CI: 2.72–27.49). Additionally, living in large families was linked to a higher intake of snack foods (AOR: 5.71, 95%CI: 1.03–31.67). Conversely, only 29% of those who frequently consumed breakfast had a lower likelihood of consuming snacks (AOR: 0.29, 95%CI: 0.09–0.92). Higher daily pocket money was associated with lower consumption of snacks (AOR: 0.20, 95%CI: 0.04–0.93) but was linked to an increased intake of processed foods (AOR: 11.47, 95%CI: 2.29, 57.50). Furthermore, being a smoker was associated with a higher likelihood of consuming snacks (AOR: 10.05, 95%CI: 1.10–91.33), and living in a large family was linked to higher consumption of snacks (AOR: 6.37, 95%CI: 1.29–31.43) (Table 4).

## 4. Discussion

This study found that the prevalence of underweight individuals was significantly high at 25.7% among adolescents in southern Thailand, while characteristics falling into the categories of overweight and obesity were observed in 14.7% and 5.8% of the population, respectively. Unhealthy dietary patterns, particularly the consumption of snacks and processed foods, were linked to various sociodemographic factors.

For underweight adolescents, those in the younger age group with higher allowances tended to consume more snacks. Males who frequently had breakfast consumed fewer processed foods. However, frequent exercise was associated with higher processed foods consumption. Among overweight adolescents, a younger age, living in larger families, and current smoking were associated with higher snack consumption. Conversely, a higher allowance and regular breakfast intake were connected to lower snack consumption. Overweight adolescents from wealthier backgrounds and larger families were more likely to consume more processed foods.

The prevalence of overweight individuals observed in this study exceeded the rates reported in northern Thailand [18,22]. However, even higher rates of overweight individuals have been documented in Mexico [23], the USA [24], and Australia [25], with percentages of 34.90%, 31.80%, and 39.10%, respectively. In contrast, the prevalence in Asia was lower than Malaysia’s 23.7% but higher than Indonesia, Laos, the Philippines, and Timor-Leste, with rates of 15.8%, 11.1%, 9.3%, and 5.7%, respectively [26]. This elevated occurrence of the overweight may be attributed to rapid economic growth leading to shifts from rural to urban living. Lifestyle changes, such as those of adolescents opting for motorbikes instead of walking or biking to school and consuming high levels of fast food or junk food near schools, contribute to this increase in the number of overweight [27].

Furthermore, this study identified a notably higher prevalence of underweight individuals compared with earlier research conducted in Ghana [9], India [28], Poland [29], Indonesia, Laos, and the Philippines [26], where rates of 6.3%, 7.3%, 9.4%, 7.9%, 5.3%, and 10.3% were reported, respectively. A similar prevalence of the underweight was observed in Timor-Leste, with a rate of 21.0% [26]. The heightened majority of underweight adolescents may be linked to cultural attitudes favoring thinness. Additionally, dietary habits in different regions of Thailand likely influence nutritional status. For instance, Southern adolescents engaged in frequent evening play after school tend to consume high-energy foods near school while skipping breakfast. In contrast, central adolescents prioritize breakfast over other meals and reduce carbohydrate intake during dinner [30]. Moreover, north eastern adolescents believe that traditional foods can help reduce the consumption of high-energy foods; these conventional foods are typically low in fat and fiber and incorporate herbs [31].

When examining sociodemographic data concerning nutritional status, significant differences were found in sex and current smoking among adolescents. The study revealed that males had a higher prevalence of the overweight than females, consistent with previous studies [32,33]. Contributing factors to weight underestimation in males included the increased consumption of unhealthy foods, as they often eat food with their friends and are more concerned about their fun than their health [34], which can lead to them becoming overweight [35]. Additionally, males who are overweight have been found to desire a smaller body shape [36], leading to dietary control in an attempt to achieve a more satisfactory body image by increasing the consumption of vegetables by more than three servings [27]. Conversely, the prevalence of the underweight was higher among males than females in this study, contrary to previous studies [32,33]. This difference could be attributed to males having lower body fat and fat mass but higher lean body mass and a higher requirement for energy and nutrients to facilitate the growth and development of lean body mass [37]. Inadequate intake of energy and nutrients could contribute to underweight status in males [38]. Being underweight during adolescence can lead to inhibited growth and development, increased prevalence of anemia, certain chronic diseases, and psychological issues such as poor self-confidence, low self-esteem, stress [2], and reproductive dysfunction [39].

Furthermore, the study found a high smoking prevalence among underweight individuals, consistent with previous studies [40]. Adolescents often believe increased smoking can help manage body weight, increasing smoking among overweight individuals [41]. Additionally, nicotine’s role is crucial in regulating smokers’ appetites by suppressing the hypothalamus [42].

The findings of this study reveal that the healthy pattern observed among Thai adolescents referred to as the ‘protein and vegetables’ or ‘traditional’ pattern, which includes foods such as milk products, soy milk, stir-fried meat, eggs, beans, pork, and fresh vegetables, accounted for the highest percentage of variance (12.498%) among the three identifies DPs. These results are consistent with previous studies [43,44], as these food items are rich sources of fiber, essential amino acids, iron, vitamin B12, zinc, and selenium [45]. Despite the overall shift in nutrition and dietary habits towards increasing consumption of fast food, ready-to-eat food, or Western-style food while reducing the intake of fresh or local vegetables among adolescents [46], it was found that many Thai adolescents still include protein and vegetable patterns in their breakfast. This may be attributed to the easy availability of these foods at roadside shops or grocery stores near the school. Additionally, traditional Thai dishes, including popular southern local Thai foods such as stir-fried meat with vegetables, Kang Som (sour curry with fish and vegetables), and Kang Liang (vegetable soup), often incorporate fresh local vegetables such as cucumber, cabbage, and morning glory [47,48].

The unhealthy DPs observed among Thai adolescents are characterized by the consumption of ‘snacks’ and ‘processed food’. Snacks are small foods or drinks typically consumed between main meals, providing high energy but low nutritional value. These snacks are often called ‘empty’ or ‘junk food’ because they lack dietary benefits [9,10]. A study on Thai adolescents revealed that their snack patterns consist of Thai desserts, snacks, and bakery products high in sugar, salt, and fat. These foods are easily accessible in school canteens and cooperatives and are popular during break time before lunch. It is noteworthy that adolescents who skip breakfast have been found to have a 71% higher risk of consuming snack foods and inadequate energy intake [49]. This can negatively impact their activity levels and academic performance during classes. Underweight adolescents, those of a younger age (13–15 years), and those who receive higher daily pocket money (more than 100 THB/day) tend to consume snacks more frequently. Previous studies have indicated that younger adolescents are less concerned about their body shape and tend to consume more energy than older adolescents [50]. Additionally, having snacks or fast foods high in salt and sodium more than twice weekly is associated with being overweight or obese [28].

Younger adolescents are less concerned with their body shape and energy intake than older adolescents, related to their increased snack consumption [50]. Additionally, these snack foods are often more expensive and palatable [50], and adolescents with higher daily pocket money have greater purchasing power, making them 3.43 times more likely to choose such options. Our study also found that overweight adolescents from large families (more than five members) were 5.71 times more likely to consume more snack foods, consistent with prior research [51]. This could be because large families may purchase unhealthy food outside the home, offering various food options at each meal [52,53]. Additionally, family structure is often associated with economic status and income, which can impact the time constraints for food preparation and the frequency of meals.

Interestingly, it was only smoking that was found to impose a 10.02 times higher likelihood of consuming snack patterns than non-smokers. This finding is consistent with other studies showing that many adolescents believe smoking can help control body weight [41]. Nicotine can affect their taste perception, changing their eating behavior [54]. Secondly, ‘processed foods’ are food products that undergo various processing techniques to enhance shelf life and taste. These foods often contain added sugar, oil, salt, and culinary components. Among Thai adolescents, popular processed food choices include instant noodles, processed meat, soda drinks, noodles, and chicken. These foods are available in school canteens and stores near their school and are commonly consumed by people before they go home in the evening. Previous researchers have also referred to these foods as ‘Western’ [11,55]. The study found that underweight adolescents, particularly males, consume processed foods less frequently than females. This is likely because females are more likely to be dissatisfied with their bodies and diet restrictively, which can lead them to eat more unhealthy energy-dense food rather than healthy food [56].

Furthermore, the study revealed that underweight individuals who exercise consistently are 2.15 times more likely to consume processed foods. They believe increasing exercise or food consumption could contribute to weight gain [57]. In addition, most adolescents participate in outdoor activities at school during the evening. This context provides ample opportunities for them to access processed foods, particularly those available near the school premises, such as food trucks or street vendors. On the other hand, obese adolescents who receive higher daily pocket money are 11.47 times more likely to consume processed foods. This finding is consistent with previous studies and is associated with higher family income [9,57]. The average allowance was revealed to be 100 THB per day, consistent with previous studies. Thai adolescents spent more than 30 THB on the intake of unhealthy foods.

Moreover, family members associated with food choices may provide suggestions and role models for diet behavior [58]. Additionally, the study found that obese adolescents from large families have a 6.37 times higher likelihood of consuming processed foods. This could be attributed to the frequency of meals and the quantity of food consumed within these families [59]. Furthermore, parents in these families may need more time to prepare meals due to work commitments.

### Limitations and Strengths of the Study

In terms of limitations, this study utilized a self-report food frequency questionnaire (FFQ) to assess dietary intake over the past month. This may lead to the under- or over-reporting of food intake compared with actual consumption. However, it is essential to note that this limitation does not diminish the significance of the study’s findings. This research provides valuable evidence-based insights into dietary patterns and sociodemographic determinants among Thai adolescents. The results have practical implications for the development of guidelines to prevent and manage weight-related issues among school-going youth.

## 5. Conclusions

The study concludes that addressing sociodemographic factors, family structure, and lifestyle behaviors is crucial when addressing unhealthy dietary patterns (DPs) among adolescents in southern Thailand. Parents, health professionals, and schools need to prioritize this issue and work together on healthier eating habits among adolescents. The study found that younger age and higher daily pocket money were significant factors associated with snack consumption among underweight adolescents. Overweight adolescents living in large families, engaging in current smoking, and skipping breakfast were more likely to consume unhealthy snacks. Similarly, underweight females who engaged in frequent exercise and overweight adolescents who received higher daily pocket money and lived in large families were more likely to consume processed foods.

The study recommends implementing healthy diet policies in schools, such as promoting healthy food, developing healthy cooking skills among vendors in the school canteen, and improving multisectoral and stakeholder participation and collaboration to provide healthy foods near the school. Because unhealthy food options are prevalent and easily accessible to adolescents, schools are crucial in promoting more nutritious food choices. Nutritional interventions and education programs should be developed to improve nutritional knowledge and promote healthy lifestyle practices among adolescents through nutrition literacy development, nutritional knowledge application in the classroom, and collaboration with families to promote healthy dietary practices. By prioritizing the dietary habits of underweight and overweight adolescents and by providing high-quality meals, schools can significantly contribute to the improvement of the nutritional health statuses of these individuals.

## Figures and Tables

**Figure 1 ijerph-20-06703-f001:**
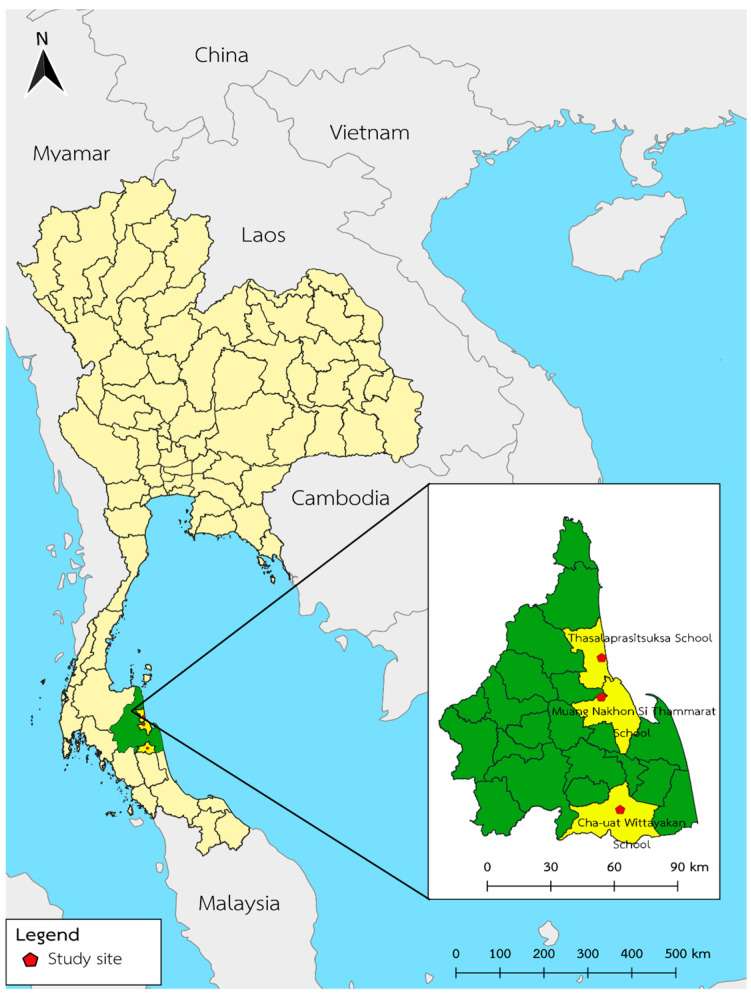
Location of the study site: Thasalaprasitsuksa school (Tha Sala district), Muang Nakhon Si Thammarat school (Muang Nakhon Si Thammarat district), and Cha-uat Wittayakan school (Cha-uat district), Nakhon Si Thammarat province, Thailand. Quantum GIS version 3.16.11 (ESRI base maps) generated the map.

**Figure 2 ijerph-20-06703-f002:**
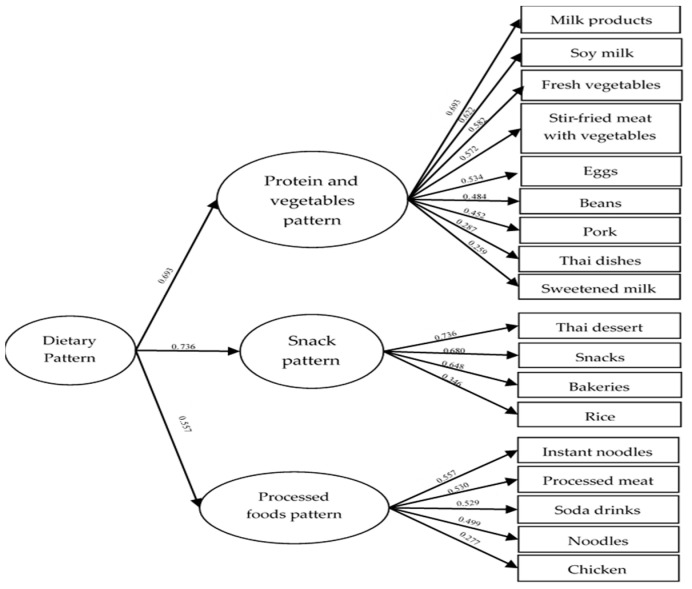
Components and factor loading of dietary patterns from principal component analysis of FFQ among Thai adolescents. Explained variances (%) of protein and vegetable pattern, snack pattern, and processed foods pattern were 12.498, 10.732, and 7.078, respectively, while eigenvalues showed respective variances of 2.626, 2.252, and 1.486. The accumulated explained variance was 30.299%.

**Table 1 ijerph-20-06703-t001:** Food groups used in the dietary pattern analysis.

Food Groups	Food Items in Each Group
Rice	Rice, sticky rice, and boiled rice
Instant noodles	Instant noodles
Noodles	Noodles with soup and rice noodles with curry
Snacks	Potato chips and crispy snacks
Bakeries	Sandwich, doughnuts, cakes, and wafers
Sweetened drinks	Iced tea/green tea and iced flavoured fruit drinks
Soda drinks	Coca-Cola, Pepsi, and other soda drinks
Thai Desserts	Fruit or flour with sugar and coconut milk
Fish	Fish in curry, fried fish, and boiled fish
Chicken	Fried chicken and grilled chicken
Pork	Grilled pork, fried pork, and cooked pork
Eggs	Fried egg, omelet, and boiled egg
Sweetened milk	Flavoured milk such as chocolate, strawberry, and coffee
Milk products	Fermented milk and yogurt
Soy milk	Soy milk
Beans	Fried peanut, boiled soybean, and green bean with sugar/coconut milk
Processed meat	Processed meat high in salt and fat, that is, Thai style Thai-style-water fish, meatballs, and bacon
Stir-fried meat with vegetables	Stir-fried meat such as chicken, pork, and shrimp with sliced vegetables
Thai dish	Curry Thai style with fish/shrimp and some vegetables, sour and spicy soup, and stir-fried vegetables
Fresh vegetables	Fresh vegetables that is, cucumbers, lentils, kale, cabbage, gourd, Thai morning glory, and green onion
Fruits	Banana, ripe mango, ripe papaya, pineapple, watermelon, and rambutan

**Table 2 ijerph-20-06703-t002:** Sociodemographic data of Thai adolescents by nutritional status.

Sociodemographic Data	Total	Nutritional Status			*p*-Value
		Underweight 364 (25.7%)	Normal 761 (53.8%)	Overweight 290 (20.5%)	
Sex	1415				0.009
Male	599 (42.3)	167 (27.9)	294 (49.1)	138 (23.0)	
Female	816 (57.7)	197 (24.1)	467 (57.3)	152 (18.6)	
Age (years)	1415				0.334
13–15	887 (62.7)	221 (24.9)	474 (53.5)	192 (21.6)	
16–18	528 (37.3)	143 (27.1)	287 (54.3)	98 (18.6)	
GPA	1174				0.421
≤3.00	742 (63.2)	183 (24.7)	410 (55.3)	149 (20.0)	
>3.00	432 (36.8)	120 (27.8)	223 (51.6)	89 (20.6)	
Daily pocket money	1415				0.771
≤100 THB per day	1171 (82.8)	301 (25.7)	626 (53.5)	244 (20.8)	
>100 THB per day	244 (17.2)	63 (25.8)	135 (55.3)	46 (18.9)	
Number of family members	1404				0.918
Small (≤5 members)	1136 (80.9)	291 (25.6)	613 (54.0)	232 (20.4)	
Large (>5 members)	268 (19.1)	70 (26.1)	141 (52.6)	57 (21.3)	
Student’s guardian	1404				0.500
Parents	939 (66.9)	239 (25.5)	499 (53.1)	201 (21.4)	
Grandparents	465 (33.1)	122 (26.2)	256 (55.1)	87 (18.7)	
Family cook	1369				
Mother	1269 (92.7)	323 (25.5)	677 (53.3)	269 (21.2)	
Grandparents/others	100 (7.3)	29 (29.0)	54 (54.0)	17 (17.0)	
Current smoking	1415				0.013
No	1232 (87.1)	301 (24.4)	677 (55.0)	254 (20.6)	
Yes	183 (12.9)	63 (34.4)	84 (45.9)	36 (19.7)	
Exercise	1415				0.716
≤3 days/week	491 (34.7)	124 (25.3)	264 (53.8)	103 (21.0)	
>3 days/week	924 (65.3)	240 (26.0)	497 (53.8)	187 (20.2)	
Breakfast intake	1415				0.928
≤3 days/week	924 (65.3)	234 (25.3)	495 (53.6)	195 (21.1)	
>3 days/week	491 (34.7)	130 (26.5)	266 (54.2)	95 (19.3)	

**Table 3 ijerph-20-06703-t003:** Multinomial logistic regression predicts risk factors of dietary patterns across quartiles among underweight adolescents.

Associated Factors	Snack Pattern AOR (95%CI)			Processed Foods AOR (95%CI)		
	Q2 (*n* = 91)	Q3 (*n* = 91)	Q4 (*n* = 91)	Q2 (*n* = 91)	Q3 (*n* = 91)	Q4 (*n* = 91)
Sex						
Male	1.29 (0.61–2.72)	1.24 (0.59–2.59)	0.74 (0.34–1.60)	0.73 (0.35–1.54)	0.25 ** (0.12–0.54)	0.22 ** (0.10–0.47)
Female	1	1	1	1	1	1
Age						
13–15 years	1.03 (0.51–2.07)	1.74 (0.87–3.48)	3.24 * (1.55–6.76)	1.32 (0.65–2.66)	1.07 (0.54–2.14)	1.86 (0.91–3.78)
16–18 years	1	1	1	1	1	1
Daily pocket money						
>100 THB/day	2.17 (0.74–6.35)	3.43 * (1.25–9.39)	2.75 (0.96–7.86)	1.45 (0.57–3.63)	0.83 (0.31–2.20)	2.03 (0.84–4.91)
≤100 THB/day	1	1	1	1	1	1
Exercise						
>3 days/week	0.80 (0.38–1.66)	1.13 (0.55–2.30)	0.90 (0.42–1.90)	1.14 (0.54–2.40)	2.15 * (1.04–4.46)	1.26 (0.5–2.71)
<3 days/week	1	1	1	1	1	1
Breakfast intake						
>3 days/week	0.98 (0.48–2.02)	1.06 (0.51–2.18)	1.35 (0.64–2.87)	0.37 * (0.17–0.81)	0.83 (0.37–1.83)	0.48 (0.22–1.05)
<3 days/week	1	1	1	1	1	1

* *p*-value < 0.05, ** *p*-value < 0.001 Q1 as a reference compared with Q2–Q4. AOR, adjusted odds ratio; CI, confidence interval. Independent variables: sex, age, GPA, daily pocket money, number of family members, primary caregiver, family cook, current smoking, exercise, and breakfast intake.

**Table 4 ijerph-20-06703-t004:** Multinomial logistic regression predicts risk factors of dietary patterns across quartiles among overweight adolescents.

Associated Factors	Snack Pattern AOR (95%CI)			Processed Foods AOR (95%CI)		
	Q2 (*n* = 91)	Q3 (*n* = 91)	Q4 (*n* = 91)	Q2 (*n* = 91)	Q3 (*n* = 91)	Q4 (*n* = 91)
Age						
13–15 years	3.15 * (1.18–8.36)	2.18 (0.82–5.79)	8.65 ** (2.72–27.49)	1.85 (0.67–5.10)	1.96 (0.72–5.30)	2.08 (0.78–5.51)
16–18 years	1	1	1	1	1	1
Daily pocket money						
>100 THB/day	0.31 (0.06–1.51)	0.20 * (0.04–0.93)	0.22 (0.04–1.00)	10.02 * (1.94–51.65)	11.47 * (2.29–57.50)	1.91 (0.66–5.52)
≤100 THB/day	1	1	1	1	1	1
Number of family members						
Large family	3.52 (0.89–13.91)	5.71 * (1.03–31.67)	0.59 (0.19–1.78)	6.37 * (1.29–31.43)	2.13 (0.54–8.32)	0.71 (0.22–2.24)
Small family	1	1	1	1	1	1
Current smoking						
Yes	10.05 * (1.10–91.33)	5.52 (0.96–31.71)	1.46 (0.36–5.89)	0.85 (0.18–3.91)	1.14 (0.24–5.38)	1.88 (0.35–9.91)
No	1	1	1	1	1	1
Breakfast intake						
>3 days/week	0.59 (0.21–1.61)	0.29 * (0.09–0.92)	0.82 (0.30–2.20)	2.86 (0.92–8.90)	1.85 (0.60–5.68)	1.90 (0.64–5.68)
<3 days/week	1	1	1	1	1	1

* *p*-value < 0.05, ** *p*-value < 0.001 Q1 as a reference compared with Q2–Q4. AOR, adjusted odds ratio; CI, confidence interval. Independent variables: sex, age, GPA, daily pocket money, number of family members, primary caregiver, family cook, current smoking, exercise, and breakfast intake.

## Data Availability

Data will be available from the corresponding author upon reasonable request, as we are not considering publicizing the raw data.

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
