# Peer review of "Unhealthy Dietary Patterns and Their Associations with Sociodemographic Factors as Predictors among Underweight and Overweight Adolescents in Southern Thailand"

_ijerph, 2023, doi:10.3390/ijerph20176703_

Round 1

Reviewer 1 Report

Reviewer Report: ijerph-2565168

 Title: Unhealthy Dietary Patterns and Their Associations with Sociodemographic Factors as Predictors among Underweight and Overweight Adolescents in Southern Thailand

Dear authors,

Congratulations on your submitted manuscript. The study of eating patterns and their relationship with sociodemographic factors among adolescents is an important topic in times of unhealthy lifestyles. The high prevalence of underweight is one of the main findings of the study. The study identified healthy (high protein and vegetable) and unhealthy (snack and processed foods) dietary patterns, with an emphasis on the influences of age, gender, and family size. The manuscript contributes to the understanding of the complex social and behavioral influences on adolescents' eating habits. However, the study could benefit from a few improvements, which I have highlighted below. My suggestions will be forwarded to the Editor.

Best regards,

The reviewer

---------------------------------------------------------------------------------------------------------------------------

Title and Abstract

- Although the summary highlights the importance of diet for adolescents, a brief justification is needed as to why the investigation of dietary patterns is crucial in adolescence;

- Please add information about sample size;

- Would it be possible to include descriptive statistics in the following sentence? “Three DPs were identified: a healthy ‘protein and vegetables’ and two unhealthy patterns: ‘snacks’ and ‘processed foods.’”

Introduction

- Overall, the introduction lays a solid foundation, but could be improved upon by providing a more detailed rationale and highlighting the importance of the study more comprehensively. Why is this particular study necessary?

- Mentioning Thailand's specific concerns regarding adolescent eating habits is certainly an important point. However, the authors could make a broader connection to the global health challenges faced by young people due to unhealthy diets.

- When mentioning statistical data such as prevalence rates of overweight and obesity in Thai adolescents, it is important to cite the sources/references of these data for greater credibility. Please provide references for the following sentences:

a) “During adolescence, a critical phase characterized by rapid growth and development, individuals have increased nutritional requirements, making them more susceptible to malnutrition.”

b) “Over the past two decades, Thai adolescents have shifted from traditional rice and vegetable-based diets to unhealthy eating patterns characterized by high fat, salt, and sugar but low fiber content.”

Methods

- Could the authors provide a brief rationale for choosing the three specific public schools and why these districts were selected to represent the different regions?

- Please consider adding a brief comment as to why you exclude sick or injured students, as this may affect nutritional status and eating habits.

- The explanation on how to calculate the sample size was well described.

- Some definitions may be useful: How did the authors arrive at the grade point average (GPA)? How was the conversion of the amount in THB to USD done?

- Nutritional status categories have been clearly defined. However, it seems to me that there is a lack of details, or better presentation of information about SD. For example, adolescents with DS between -1 and -2 were classified in what way?

- What additional information could the authors provide about the adaptation and validation of the FFQ?

- It may be beneficial to clarify that the FFQ response scale ranges from "never" to "more than five times a week" to help readers understand the scale.

- Explaining in a more simplified way what PCA is and why it was chosen can help readers who are not familiar with this technique.

Results

- In my opinion, the data are adequately presented and follow a logical order following the purposes of the study. However, the tables are disorganized and need adjustments. Furthermore, Figure 2 is not of good quality. Please try to improve the visual aspects of Figure 2.

Discussion

- In the first paragraph, make it clear what practical implications these results have. The discussion could be organized more clearly and structured. The authors could begin by briefly summarizing the main findings (including associated factors), highlighting the prevalence of underweight, overweight, and obesity. What are the practical implications of these findings for society and public health? The prevalence of underweight found in this study is an important finding and should be highlighted. By recognizing that more than 25% of the population have underweight, it will be possible to highlight the need for awareness to reverse this trend;

- Were the rates of overweight, obesity, and underweight compared with data from other similar countries?

- When authors mention that overweight and obesity rates are higher than in some regions and countries, they should consider exploring possible reasons for these differences;

- How might differences be influenced by cultural, economic, and lifestyle factors? Present objective information;

- When discussing differences in sociodemographic factors and nutritional status, it is important to go beyond simply identifying differences and explore possible underlying mechanisms. For example, why do men have a higher prevalence of overweight and why might this be related to unhealthy food consumption?

- After identifying differences in overweight, obesity, and underweight patterns between genders, it would be interesting to discuss the implications of these results in terms of public health and intervention actions. If you find it interesting, objectively add the implications in the study conclusions;

- Please explain objectively how frequent consumption of snacks and processed foods can affect health, including risks of obesity, chronic diseases, and nutritional deficiencies.

- The authors mention that younger teens tend to consume more snacks - why? Also, discuss the implications of gender and family size differences in choosing these foods.

- Explain the relationship between allowance and food choices. Furthermore, discuss whether this association might indicate a possible role for family income in influencing food choices.

- How might exercise be related to the consumption of processed foods? It would be important to elaborate a little more on the reasoning behind this association.

Conclusions

- In addition to food policies, highlight the importance of nutrition education in schools. Objectively explain how educational programs can empower teens to make informed, healthy food choices throughout their lives.

References

Please verify the references and confirm whether the journal titles should be italicized or not, following the author's guidelines. The year of publication should be in bold.

Author Response

Response to reviewer1's comments

Title and Abstract

- Although the summary highlights the importance of diet for adolescents, a brief justification is needed as to why the investigation of dietary patterns is crucial in adolescence;

Answer: I have stated this on page 1, L18-19.

- Please add information about sample size;

Answer: I have added more details on page 1, L21-22.

- Would it be possible to include descriptive statistics in the following sentence? "Three DPs were identified: a healthy' protein and vegetables' and two unhealthy patterns: 'snacks' and 'processed foods.'

Answer: I included descriptive statistics on page 1, L26-27.

Introduction

- Overall, the introduction lays a solid foundation but could be improved upon by providing a more detailed rationale and highlighting the importance of the study more comprehensively.

Why is this particular study necessary?

Answer: I provided a more detailed rationale on pages 1-2, L39-47.

- Mentioning Thailand's specific concerns regarding adolescent eating habits is an important point. However, the authors could make a broader connection to the global health challenges young people face due to unhealthy diets.

Answer: I have connected to WHO's goal to reduce the prevalence of obese adolescents on page 2, L55-58.

- When mentioning statistical data such as prevalence rates of overweight and obesity in Thai adolescents, it is essential to cite the sources/references of these data for greater credibility. Please provide references for the following sentences:

  1. a) "During adolescence, a critical phase characterized by rapid growth and development,

individuals have increased nutritional requirements, making them more susceptible to

malnutrition."

Answer: I have cited the reference [1] on page 1, L41.

  1. b) "Over the past two decades, Thai adolescents have shifted from traditional rice and

vegetable-based diets to unhealthy eating patterns characterized by high fat, salt, and sugar

but low fiber content."

Answer: I have cited the reference [16] on page 2, L77.

Methods

- Could the authors provide a brief rationale for choosing the three specific public schools

and why these districts were selected to represent the different regions.

Answer: I have provided a brief rationale for choosing the three schools on page 2, L91, 94-95.

- Please consider briefly commenting on why you exclude sick or injured students, as

this may affect nutritional status and eating habits.

Answer: I have added a reason why exclude sick or injured students on page 3, L102.

- The explanation of how to calculate the sample size was well described.

Answer: Thank you.

- Some definitions may be helpful to: How did the authors determine the grade point average (GPA)?

Answer: I have explained on page 4, L150-151.

How was the conversion of the amount in THB to USD done?

Answer: I have added the exchange rate on page 4, L153.

- Nutritional status categories have been clearly defined. However, there is a lack of details or a better presentation of information about SD.

Answer: I have added the information about SD on page 4, L169-175.

- What additional information could the authors provide about the adaptation and validation

of the FFQ?

Answer: I have added information for the adaptation on page 4, L179-180, and the validation on L191-192.

- It may be beneficial to clarify that the FFQ response scale ranges from "never" to "more than five times a week" to help readers understand the scale.

Answer: I have clarified the FFQ response scale ranges on page 4, L188-190.

- Explaining what PCA is and why it was chosen can help readers unfamiliar with this technique.

Answer: I have explained what PCA is and why it was chosen on page 5, L201-205

Results

-The data are adequately presented and follow a logical order following the purposes of the study. However, Figure 2 is not of good quality. Please try to improve the visual aspects of Figure 2.

Answer: I have improved the visual aspects of Figure 2 on page 7.

Discussion

- In the first paragraph, clarify what practical implications these results have. The discussion could be organized more clearly and structured. The authors could begin by briefly summarizing the main findings (including associated factors), highlighting the prevalence of underweight, overweight, and obesity. What are the practical implications of these findings for society and public health? The prevalence of underweight found in this study is an important finding and should be highlighted.

Answer: I have summarized the principal finding on page 9, L335-347.

By recognizing that more than 25% of the population have underweight, it will be possible to highlight the need for awareness to reverse this trend;

- Were the rates of overweight, obesity, and underweight compared with data from other similar countries?

Answer: I have compared data to other countries on page 9, L348-362.

- When authors mention that overweight and obesity rates are higher than in some regions and countries, they should consider exploring possible reasons for these differences;

Answer: I have explored the reasons on pages 10-11, L362-369.

- When discussing differences in sociodemographic factors and nutritional status, it is essential to go beyond simply identifying differences and explore possible underlying mechanisms. For example, why do men have a higher prevalence of overweight, and why might this be related to unhealthy food consumption?

Answer: I have explained more on page 10, L375-381.

- After identifying differences in overweight, obesity, and underweight patterns between genders, it would be interesting to discuss the implications of these results in terms of public health and intervention actions. If you find it interesting, objectively add the impact in the study conclusions;

Answer: I have added implications in the conclusion on page 12, L487-494.

- Please explain objectively how frequent consumption of snacks and processed foods can affect health, including risks of obesity, chronic diseases, and nutritional deficiencies.

Answer: I have explained more on page 11, L422-424.

- Why do the authors mention that younger teens consume more snacks?

Answer: I have explained the reasons on page 11, L425-426.

Also, discuss the implications of gender and family size differences in choosing these foods.

Answer: I have explained this on page 11, L429-434.

- Explain the relationship between allowance and food choices. Furthermore, discuss whether this association might indicate a possible role for family income in influencing food choices.

Answer: I have explained this on page 11, L457-458.

- How might exercise be related to the consumption of processed foods? It would be essential to elaborate more on this association's reasoning.

Answer: I have explained this on page 11, L449-454.

Conclusions

- In addition to food policies, highlight the importance of nutrition education in schools.

Objectively explain how educational programs can empower teens to make informed, healthy

food choices throughout their lives.

Answer: I have highlighted this on page 12, L487-494.

References

Please verify the references and confirm whether the journal titles should be italicized,

following the author's guidelines. The year of publication should be in bold.

Answer: I have revised all references on pages 13-15.

Reviewer 2 Report

Comments and suggestions

It was my pleasure to review this manuscript, which focuses on examining unhealthy dietary patterns and their potential associations with sociodemographic factors among underweight and overweight adolescents in southern Thailand. The primary objective of this study is to assess the dietary patterns of Thai adolescents and explore the sociodemographic variables linked to unhealthy dietary patterns among those who are underweight or overweight. This cross-sectional study was conducted across three districts representing the northern, central, and southern regions of Nakhon Si Thammarat province. The questionnaire survey was carried out between January and April 2021, involving a sample of 1415 participants.

In summary, this manuscript offers a comprehensive and in-depth exploration of the adoption process. I found the subject matter to be particularly intriguing, and the manuscript itself is well-organized, presenting information in a coherent manner. Nevertheless, there are some errors that have surfaced in the results section. Therefore, I kindly urge you to meticulously review the contents prior to submitting the revised version of the manuscript. With the primary aim of enhancing the manuscript's quality, I would like to provide several comments.

Introduction:

The introduction presents a substantial description that underscores the role of easy access to street food, such as snacks, sugary drinks, and high-energy-dense items sold at tuck shops and convenience stores near schools or along school routes. This assertion is grounded in a previous survey conducted by the Health System Research Institute in Bangkok, Thailand. Consequently, the significance of maintaining a healthy dietary intake from adolescence to adulthood is highlighted as pivotal in perpetuating these positive behaviors, as indicated by prior research findings. This emphasis serves to underscore the necessity for conducting a similar study and offers a more precise elucidation of the innovative rationale and objectives underpinning the research. Moreover, the introduction integrates informative and contemporary references from the international literature, thereby augmenting the credibility and relevance of the study.

Materials and Methods:

Firstly, the method section is thoughtfully organized and offers crucial insights, particularly in the areas of sample size estimation, as well as the development and validation of the study instrument.

Secondly, the questionnaire meticulously gathered data on each food item, with participants indicating the frequency of their consumption over the past month. It's worth noting that this approach could potentially introduce recall bias. To its credit, this limitation has been duly acknowledged within the confines of the study.

Statistical Analysis (Page 5, lines 199-201):

You mentioned that "Multinomial logistic regression analysis was conducted to examine the relationship between potential sociodemographic variables and Q2-Q3 of the DPs, with Q1 as the reference category." It appears that regression analysis was actually performed to investigate the association between potential sociodemographic variables and Q2-Q4 of the dietary patterns based on your analysis (as demonstrated in tables 4 and 5). Please confirm the accuracy of this statement.

Page 8, lines 292-293:

The sentence "Independent variables: sex, age, GPA, daily pocket money, number of family members, primary caregiver, family cook, current smoking, exercise, and breakfast intake" should be included in Table 4.

Ensuring its placement in the appropriate table could help prevent any potential confusion among readers. Please verify this arrangement.

Page 9, lines 301-303:

Your contents mentioned that “Furthermore, being a smoker was associated with a lower likelihood of consuming snacks (AOR: 10.05, 95%CI: 1.10-91.33), and living in a large family was linked to lower consumption of snacks (AOR: 6.37, 95%CI: 1.29-31.43) (table 5).” The correct interpretation should indeed be: "Furthermore, being a smoker was associated with a higher likelihood of consuming snacks (AOR: 10.05, 95%CI: 1.10-91.33), and living in a large family was linked to higher consumption of snacks (AOR: 6.37, 95%CI: 1.29-31.43) (table 5)." Please confirm the accuracy of this statement.

Discussion:

Page 10, lines 363-365:

The content states that "Younger adolescents (13-15 years) who are overweight or underweight and receive higher daily pocket money (more than 100 THB/day) tend to consume snacks more frequently." However, table 5 actually reveals that higher daily pocket money was associated with lower consumption of snacks (AOR: 0.20, 95%CI: 0.04-0.93). Please confirm this correction.

Page 11, lines 376-377:

The content states that “only smoking was found to have a 10.02 times lower likelihood of consuming snack patterns than nonsmokers.” In fact, this sentence should states only smoking was found to have a 10.02 times higher likelihood of consuming snack patterns than nonsmokers. Please confirm this correction.

Page 11, lines 386-387:

You mentioned that “The study found that underweight adolescents, particularly females, consume processed foods less frequently, accounting for 78% of this group.” In the current manuscript's results section, the provided information is not included. I recommend either incorporating the relevant table into the manuscript or adding an explanatory note to this sentence, such as "data not shown in the table." This approach would provide greater clarification. Please confirm this suggestion.

Conclusions:

Page 11, lines 411-412:

Please confirm the results of this sentence “The study found that younger age and higher daily pocket money were significant factors associated with snack consumption among underweight and overweight adolescents”.

Page 11, lines 415-417:

"Underweight females who engaged in frequent exercise and overweight adolescents who received higher daily pocket money and lived in large families were reported to have a higher likelihood of consuming processed foods."

To clarify and reinforce this finding, I suggest adding a table to your manuscript to present the relevant data. This would help substantiate the conclusion and provide readers with direct evidence of the associations you mentioned.

Author Response

Response to reviewer2's commments

Introduction:

The introduction presents a substantial description that underscores the role of easy access to street food, such as snacks, sugary drinks, and high-energy-dense items sold at tuck shops and convenience stores near schools or along school routes. This assertion is grounded in a previous survey conducted by the Health Systems Research Institute in Bangkok, Thailand. Consequently, the significance of maintaining a healthy dietary intake from adolescence to adulthood is highlighted as pivotal in perpetuating these positive behaviors, as indicated by prior research findings. This emphasis underscores the necessity for conducting a similar study and offers a more precise elucidation of the research's innovative rationale and objectives. Moreover, the introduction integrates informative and contemporary references from the international literature, thereby augmenting the credibility and relevance of the study.

Answer: Thank you so much for your comments.

 Materials and Methods:

Firstly, the method section is thoughtfully organized and offers crucial insights, particularly in sample size estimation, as well as the development and validation of the study instrument.

Secondly, the questionnaire meticulously gathered data on each food item, with participants indicating the frequency of their consumption over the past month. This approach could introduce recall bias. To its credit, this limitation has been duly acknowledged within the confines of the study.

Answer: Thank you so much for your comments. Therefore, I will state this in the limitation on page 11, L467-469.

Statistical Analysis (Page 5, lines 199-201):

You mentioned, "Multinomial logistic regression analysis was conducted to examine the relationship between potential sociodemographic variables and Q2-Q3 of the DPs, with Q1 as the reference category." Regression analysis was performed to investigate the association between potential sociodemographic variables and Q2-Q4 of the dietary patterns based on your analysis (as demonstrated in Tables 3 and 4). Please confirm the accuracy of this statement.

Answer: I confirm the accuracy of this statement on page 5, L221, pages 8, 313, page 9, L 330.

Page 8, lines 292-293:

The sentence "Independent variables: sex, age, GPA, daily pocket money, number of family members, primary caregiver, family cook, current smoking, exercise, and breakfast intake" should be included in Table 3. Ensuring its placement in the appropriate table could help prevent potential confusion among readers. Please verify this arrangement. Page 9, lines 301-303:

Answer: I have added this sentence on page 8, L314-315.

Your contents mentioned that "Furthermore, being a smoker was associated with a lower likelihood of consuming snacks (AOR: 10.05, 95%CI: 1.10-91.33), and living in a large family was linked to lower consumption of snacks (AOR: 6.37, 95%CI: 1.29-31.43) (table 4)." The correct interpretation should indeed be: "Furthermore, being a smoker was associated with a higher likelihood of consuming snacks (AOR: 10.05, 95%CI: 1.10-91.33), and living in a large family was linked to higher consumption of snacks (AOR: 6.37, 95%CI: 1.29-31.43) (table 4)." Please confirm the accuracy of this statement.

Answer: Thank you so much. I have rewritten and followed your suggestion on page 8, L324-326.

Discussion:

Page 10, lines 363-365: The content states that "Younger adolescents (13-15 years) who are overweight or underweight and receive higher daily pocket money (more than 100 THB/day) tend to consume snacks more frequently." However, table 4 reveals that higher daily pocket money was associated with lower consumption of snacks (AOR: 0.20, 95%CI: 0.04-0.93). Please confirm this correction.

Answer: Page 11, lines 418-420: I have rewritten the content to be "Underweight adolescents, younger (13-15 years) and receiving higher daily pocket money (more than 100 THB/day) tend to consume snacks more frequently," as revealed in Table 3 of underweight adolescents (page 8 L291-294).

Page 11, lines 376-377: The content states that "only smoking was found to have a 10.02 times lower likelihood of consuming snack patterns than nonsmokers." This sentence should note that only smoking was found to have a 10.02 times higher likelihood of consuming snack patterns than nonsmokers. Please confirm this correction.

Answer: Thank you so much. I have rewritten and followed your suggestion on page 11, L435-436.

Page 11, lines 386-387: You mentioned, "The study found that underweight adolescents, particularly females, consume processed foods less frequently, accounting for 78% of this group." The provided information is not included in the current manuscript's results section. I recommend incorporating the relevant table into the manuscript or adding an explanatory note to this sentence, such as "data not shown in the table." This approach would provide greater clarification. Please confirm this suggestion.

Answer: Thank you so much. I have rewritten it to be "The study found that underweight adolescents, particularly males, consume processed foods less frequently than females" on page 11, L445-446, as shown in Table 3 of underweight adolescents (page 8, L295-297).

Conclusions:

Page 11, lines 411-412: Please confirm the results of this sentence "The study found that younger age and higher daily pocket money were significant factors associated with snack consumption among underweight and overweight adolescents."

Answer: I have rewritten by cutting off "overweight" on page 12, L479-481.

 Page 11, lines 415-417: "Underweight females who engaged in frequent exercise and overweight adolescents who received higher daily pocket money and lived in large families were reported to have a higher likelihood of consuming processed foods."

Answer: Thank you so much. I have rewritten and followed your suggestion on page 12, L483-485.

To clarify and reinforce this finding, I suggest adding a table to your manuscript to present the relevant data. This would help substantiate the conclusion and provide readers with direct evidence of the associations you mentioned.

Answer: Thank you so much for your suggestion to add a table, but the pages of the manuscript have limited it.

Round 2

Reviewer 1 Report

Dear Authors,

I wanted to inform you that I have carefully reviewed the improvements you made to the manuscript. I have forwarded my feedback to the editor, highlighting the positive impact of your revisions on the overall quality of the paper. Your hard work and dedication are evident, and I believe these improvements have significantly strengthened the manuscript.

Best regards,